# Early evolution of small body size in *Homo floresiensis*

Yousuke Kaifu [1] ✉, Iwan Kurniawan [2] ✉, Soichiro Mizushima[3], Junmei Sawada[4], Michael Lague[5], Ruly Setiawan[2], Indra Sutisna[6], Unggul P. Wibowo[6], Gen Suwa [1], Reiko T. Kono [7], Tomohiko Sasaki[8], Adam Brumm [9] & Gerrit D. van den Bergh [10] ✉

Recent discoveries of *Homo floresiensis* and *H. luzonensis* raise questions regarding how extreme body size reduction occurred in some extinct *Homo* species in insular environments. Previous investigations at Mata Menge, Flores Island, Indonesia, suggested that the early Middle Pleistocene ancestors of *H. floresiensis* had even smaller jaws and teeth. Here, we report additional hominin fossils from the same deposits at Mata Menge. An adult humerus is estimated to be 9–16% shorter and thinner than the type specimen of *H. floresiensis* dated to ~60,000 years ago, and is smaller than any other Plio-Pleistocene adult hominin humeri hitherto reported. The newly recovered teeth are both exceptionally small; one of them bears closer morphological similarities to early Javanese *H. erectus*. The *H. floresiensis* lineage most likely evolved from early Asian *H. erectus* and was a long-lasting lineage on Flores with markedly diminutive body size since at least ~700,000 years ago.

The So'a Basin in central Flores, Indonesia, is a key region for elucidating the origin and evolution of *H. floresiensis*, a diminutive hominin species known from the Late Pleistocene at Liang Bua, a limestone cave in western Flores[1,2]. As with another small-bodied *Homo* discovered in Luzon[3], the evolutionary history of this insular hominin species has been the subject of protracted debate[4]. Previous field studies of the Early to Middle Pleistocene (Calabrian–Chibanian) sequence of the So'a Basin have recovered fossil remains of endemic fauna (dwarfed *Stegodon*, Komodo monitor, giant rat, birds, crocodiles and tortoises)[5,6], technologically simple stone artefacts (the oldest of which date back to at least $1.02 \pm 0.02$ million years ago [Ma])[7,8], and, importantly, a fragmentary mandible and six isolated teeth of a small-sized hominin[9]. These hominin fossils were excavated from a sandstone layer of fluvial origin (Layer II) of the upper fossil-bearing interval at the Mata Menge site, which is dated to between 0.65 and 0.773 Ma[5,6]. These fossils exhibit general morphological affinities to the type series

of *H. floresiensis* from Liang Bua (0.1–0.06 Ma)[10] and to early *H. erectus* from Java (1.1–0.8 Ma)[11], but lack the unique molar specializations characterizing the former and were substantially smaller than the latter[9].

Overall, the Mata Menge fossils suggest that they represent an ancestral segment of the Liang Bua *H. floresiensis* lineage, and that the Flores hominins are dwarfed descendants of large-bodied early Asian *H. erectus*[9]. Some cladistic/phylogenetic analyzes, however, support a direct evolutionary link between *H. floresiensis* and smaller-bodied basal *Homo* such as *H. habilis* or even *Australopithecus*[12–14]. It is important to resolve this controversy in order to elucidate the pattern and timing of body size evolution in the Flores hominins.

Notably, the Mata Menge mandible and teeth are slightly smaller than the type specimens of *H. floresiensis* from Liang Bua. This suggests that drastic dentognathic reduction had occurred on Flores by the early Middle Pleistocene epoch, more than 600,000 years before the

[1]The University Museum, The University of Tokyo, Tokyo, Japan. [2]Center for Geological Survey, Geological Agency, Bandung, Indonesia. [3]Department of Anatomy, St. Marianna University School of Medicine, Kanagawa, Japan. [4]Institute of Physical Anthropology, Niigata University of Health and Welfare, Niigata, Japan. [5]School of Natural Sciences and Mathematics, Stockton University, Stockton, NJ, USA. [6]Geology Museum Bandung, Geological Agency, Bandung, Indonesia. [7]Faculty of Letters, Keio University, Kanagawa, Japan. [8]The Kyoto University Museum, Kyoto University, Kyoto, Japan. [9]Australian Research Centre for Human Evolution, Griffith University, Brisbane, QLD, Australia. [10]Centre for Archaeological Science, School of Earth, Atmospheric and Life Sciences, University of Wollongong, Wollongong, NSW, Australia. ✉e-mail: kaifu@um.u-tokyo.ac.jp; kurniawanmgb@gmail.com; gert@uow.edu.au

earliest fossil evidence for *H. floresiensis* at Liang Bua. Until now, however, the lack of postcranial elements in the Mata Menge assemblage had limited our understanding of body size evolution on Flores.

In this paper, we report the discovery and morphology of a hominin postcranial fossil from Mata Menge, an extremely small distal humeral shaft (SOA-MM9) (Fig. 1). This specimen and two small-sized teeth (SOA-MM10 and SOA-MM11) were recovered as additions to the existing hominin assemblage from Layer II at this site (Table 1). Our histomorphic examination confirms the adult status of the humerus. We also show that shaft morphology is more similar to small-bodied *Homo* (e.g., LB1 and *H. naledi*) than to *Australopithecus* (e.g., A.L. 288-1), and a molar crown (SOA-MM11) bears closer shape similarities to early Javanese *H. erectus* than to early African *Homo*. The increased Mata Menge fossil sample supports its classification to an early representative of *H. floresiensis*, which probably experienced drastic body size reduction from large-bodied Asian *H. erectus* sometime between ~1.0 and 0.7 Ma.

## Results

### Context and geological age

All hominin fossils originated from the top of a ribbon-shaped, indurated pebbly sandstone layer (Layer II), which was deposited in a small stream channel on a volcaniclastic alluvial fan[5] ~ 0.7 Ma ago. This age estimate is based on the identification of the Brunhes-Matuyama boundary[15] dated at 0.773 Ma by palaeomagnetic measurements combined with a series of fission track dates on zircons in tuffaceous strata stratigraphically 16.5 m below Layer II[5,6]. A minimum age of $0.65 \pm 0.02$ Ma for Layer II is provided by a $^{40}Ar/^{39}Ar$ date on single hornblende crystals from an airfall tephra (PGT-2) occurring stratigraphically at 14 m above Layer II.

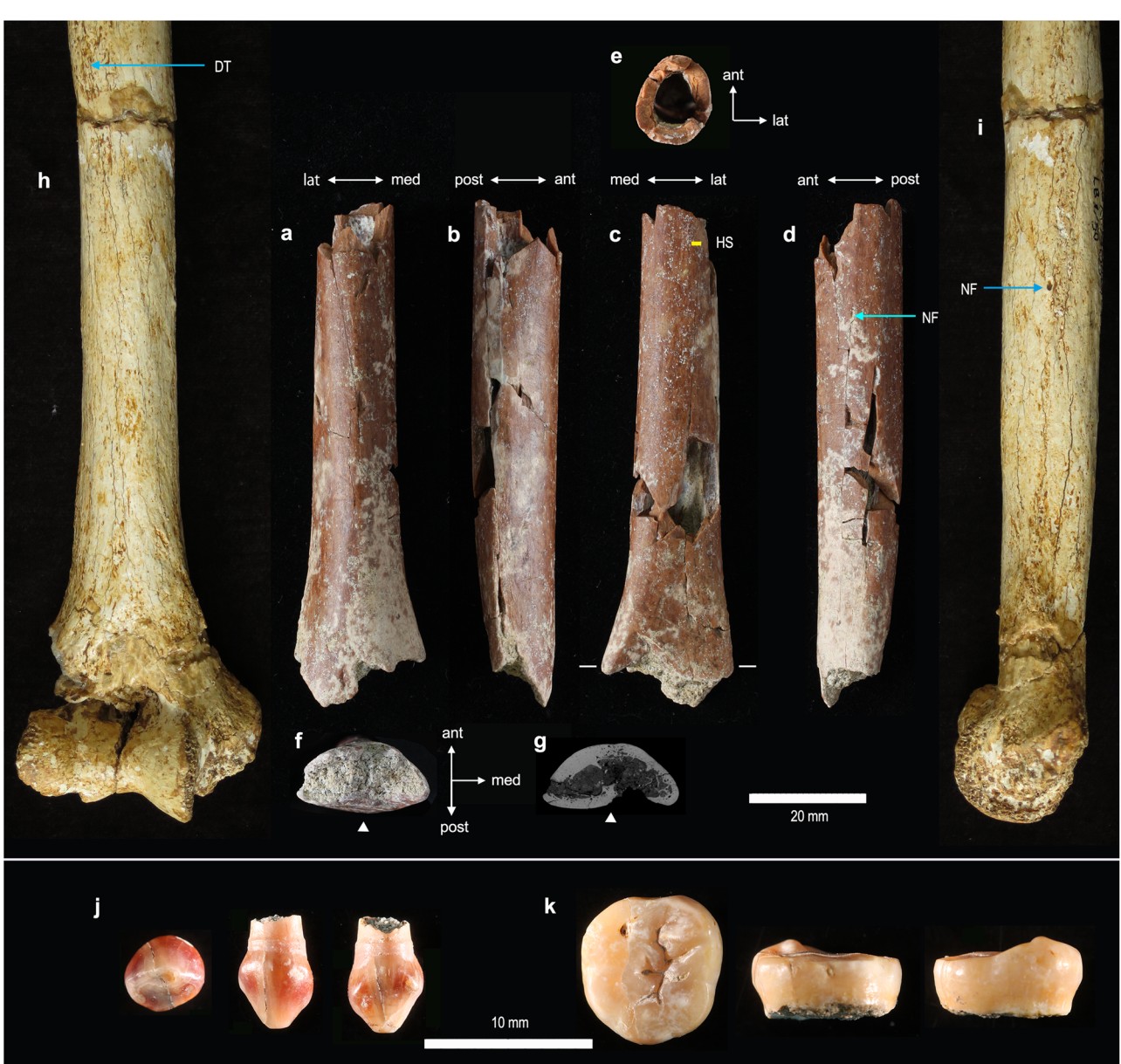

**Fig. 1 | New fossils from Mata Menge. a–f** SOA-MM9 humerus in anterior, lateral, posterior, medial, proximal, and distal views. **g** Micro-CT section of SOA-MM9 at the distal end indicated in (**c**). **h** and **i** LB1 humerus in anterior and medial views. Note the hollowed area on the posterior surface of the distal end (filled triangles in **f** and **g**), which serves as an osteometric landmark (hOF point). Abbreviations: ant = anterior, post = posterior, lat = lateral, med = medial, DT = deltoid tuberocity, NF = nutrient foramen, HS = location for histological section. **j** SOA-MM10 right d$^c$. From left to right, occlusal, labial, and lingual views. **k** SOA-MM11 left M$_3$. From left to right, occlusal, buccal, and lingual views.

**Table 1 | Hominin fossil collection from Mata Menge, So'a Basin, Flores**

| Specimen No. | Catalog No. | Discovery | Portion | Age/individual | Ref. |
|---|---|---|---|---|---|
| SOA-MM1 | MM14-T32D-F191 | 2014 Oct 08 | lt. $M_1$ (or $M_2$) | adolescent (or young adult) | 9 |
| SOA-MM2 | MM14-T32C-F234 | 2014 Oct 14 | lt. $I^1$ | adult | 9 |
| SOA-MM3 | MM14-T32D-F384 | 2014 Oct 14 | hominin cranial fragment? | | |
| SOA-MM4 | MM14-T32C-F277 | 2014 Oct 14 | rt. mandibular body | adult | 9 |
| SOA-MM5 | MM14-T32C-F452 | 2014 Oct 16 | rt. $P^3$ | adult | 9 |
| SOA-MM6 | MM14-T32B-F94 | 2014 Oct 18 | rt. $LI_{1/2}$ | adult | 9 |
| SOA-MM7 | MM14-T32C-dry sieve | 2014 Oct 21 | lt. $d_c$ | child 1† | 9 |
| SOA-MM8 | MM14-T32B-dry sieve | 2014 Oct 24 | rt. $d_c$ | child 2† | 9 |
| SOA-MM9 | MM13-T32-F159 | 2013 Oct 11* | rt. distal humerus | adult | this study |
| SOA-MM10 | MM15-T32C-F2820 | 2015 Oct 26 | rt. $d^c$ | child 1 or 3‡ | this study |
| SOA-MM11 | MM16-T32A-F4444 | 2016 April 25 | lt. $M_3$ | adult (different from SOA-MM1) | this study |

*Recognized as hominin in 2015.
†Reported as an "infant" previously[9] but a child of 3-9 years old in modern human standard[65] is more appropriate.
‡3-10 years old in modern human standard[66].

Layer II, with a maximum thickness of 50 cm, overlies a reddish paleosol (Layer III) with an undulating erosional contact. A series of massive, tuffaceous clay-rich mudflow layers (Layers Ia-f) sealed off Layers II and III subsequently (Fig. 2; Supplementary Note 1). The humerus fragment SOA-MM9 was retrieved in several pieces within one week of opening Excavation 32 A in 2013, but was recognized as such only in 2015 after reconstruction in the laboratory. The specimen was damaged in the process of excavating it from the extremely compact sandstone of Layer II. A maxillary deciduous canine ($d^c$: SOA-MM10) was excavated in 2015 at -5 cm below the boundary between Layers I and II, while a mandibular third molar ($M_3$: SOA-MM11) was excavated in 2016 at -15 cm below the top of Layer II. All hominin fossils are concentrated in the upper part of Layer II, while fossils of other fauna tend to be distributed more evenly in this unit. There is evidence for fluvial transportation of the fossils prior to burial, with many (but not all) specimens fractured (apart from excavation damage), weathered and/or rounded to some extent[16]. However, the three hominin fossils described here show minimal/no evidence of abrasion. Taphonomic and sedimentological observations suggest that the hominin fossils were deposited during a moderate to low-energy flow regime in the stream, following a relatively brief period on the surface during which the remains were disarticulated (Supplementary Note 1). Shortly after incorporation of the fossils in the stream bed the entire stream valley was filled with a 6.5 m thick sequence of mudflows. Succeeding field excavations in 2017–2019 and 2023 have yielded no more hominin fossils from this site.

## Developmental age of the humerus (SOA-MM9)

This specimen is an undistorted, distal half of the right humeral shaft that measures 88 mm in maximum preserved length (Fig. 1; Supplementary Note 2). Despite its small size, cortical bone histomorphology of SOA-MM9 clearly indicates its adult status. We examined its development stage based on age-associated increase of osteons and related structures, a method widely utilized for age estimates of extant and fossil hominins[17–21].

Histological sections were examined for cortical samples taken at the mid-posterior shafts of SOA-MM9 ('HS' in Fig. 1) and from a modern human sample (N = 20, see Supplementary Data 1). To allow for regional variation in osteon formation within each bone area[21], data were collected from two additional (nearby) sites in the midshaft for all the modern human (*H. sapiens*) specimens (Fig. 3a). In two parameters indicative of bone maturity, Osteon Population Density (OPD)[21] and Haversian Canal Index (HCI), SOA-MM9 was found to exhibit distinctly greater values (OPD = 16.5, HCI = 0.85) than in any of the modern human subadult humeri (0.0–8.9 and 0.0–0.63, respectively) (Fig. 3b, Supplementary Data 1). The values for the Mata Menge humerus are also greater than the means of our modern human adult samples (13.6 and 0.78, respectively), indicating that the SOA-MM9 individual was well within adulthood at time of death. Although the external cortical surfaces of SOA-MM9 exhibit microscopic damages that might have reduced one of the marginal osteons to half (-100 microns) (Fig. 3c), such post-depositional alterations would have limited impact on our age estimation. Even if we assume surface abrasion of 200 microns, the OPD value for SOA-MM9 would drop only slightly to -15.8. Furthermore, the dominance of secondary osteons in the outer cortex (Fig. 3c) indicates that subperiosteal bone growth during the growth period had already been terminated in this individual[22].

No evidence of pathology was found in SOA-MM9. Cortical bone thinning and woven bone would be pathognomic of some metabolic disorders[17], but these features are not evident in SOA-MM9. The relative cortical bone thickness of SOA-MM9 (0.07: the ratio of cortical bone thickness relative to the humeral shaft circumference) (Supplementary Data 1) is almost identical to the mean for the modern human adult sample (0.069). Patients with osteogenesis imperfecta, which may lead to short stature, exhibit subnormal OPD values[19], a tendency that is in opposition to the SOA-MM9 condition. Additionally, the weak but distinct development of the lateral supracondylar ridge of SOA-MM9 (Fig. 3d) indicates normal development of the extensor carpi radialis longus muscle.

## Humeral size

In all available dimensions of shaft diameter/circumference and length, SOA-MM9 is smaller than LB1 (*H. floresiensis*) and any other adult individuals of small-bodied fossil hominins (*Australopithecus* and *H. naledi*: Supplementary Data 2). Its minimum circumference (46 mm) is less than U.W. 101-283 (47.5 mm), BOU-12/1 (52 mm), and the smallest humeri in our prehistoric modern human sample (46.5 mm, N = 1050, see Supplementary Data 2). Centroid size at the -19% level cross-section is also the smallest compared to any sampled adult specimens of *Australopithecus*, *Paranthropus*, and *Homo* including *H. naledi* and Liang Bua *H. floresiensis* (Fig. 4, Supplementary Data 3). The distal shaft length measured between the NF (nutrient foramen) and hOF (superior margin of the hollow leading to the olecranon fossa) points of SOA-MM9 (58 mm) is distinctly shorter than the other hominin fossils, including LB1 (64 mm) (Supplementary Data 2), although the vertical position of NF is variable in human humeri[23].

The fragmentary nature of SOA-MM9 precludes a precise reconstruction of its original length, but it can be estimated as follows. First, the preserved proximo-posterior end of SOA-MM9 is very close to the 50% level because this portion exhibits the following suite of features characteristic of hominin humeral midshafts: 1) The radial sulcus (spiral groove) is present not on the

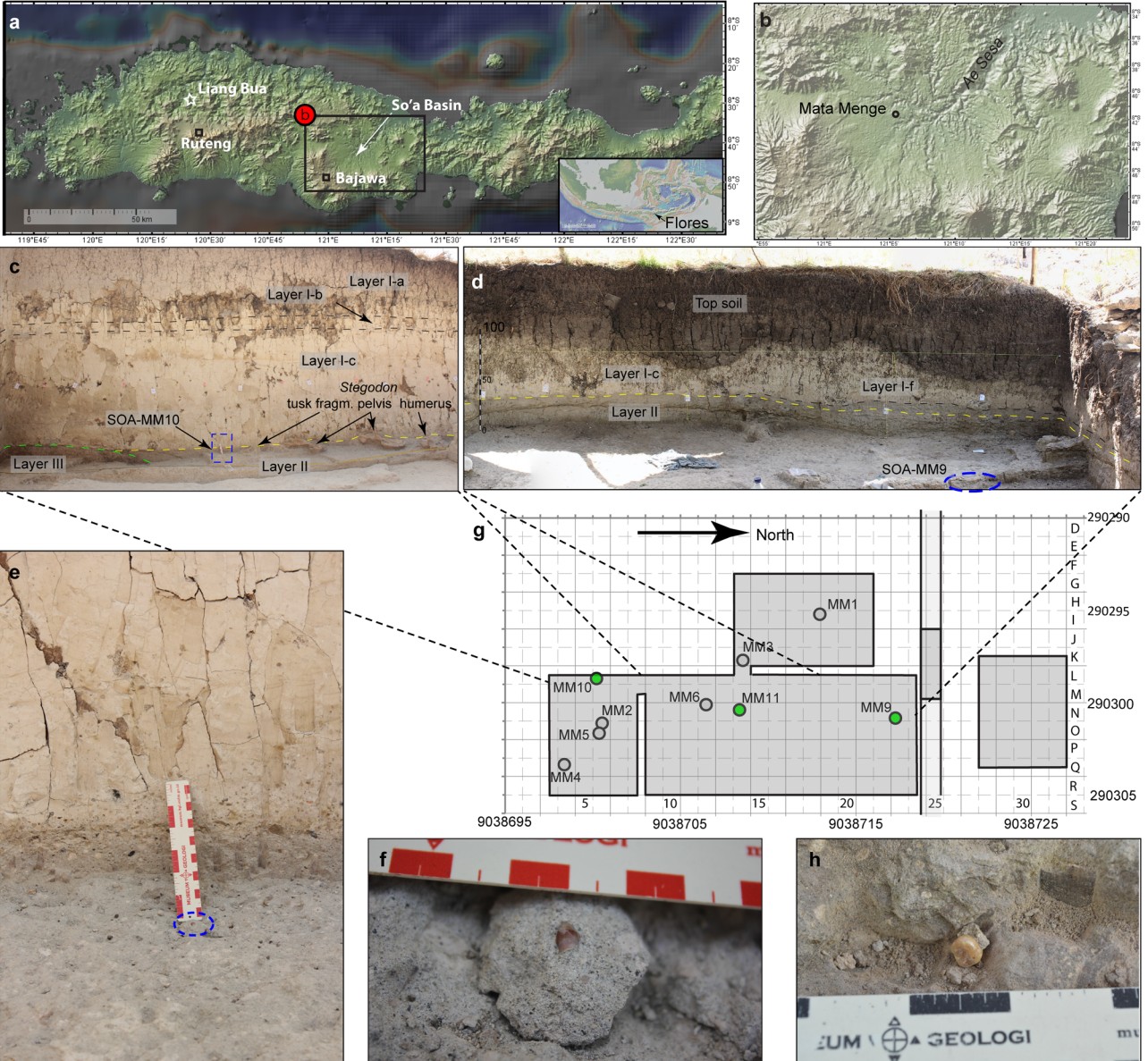

**Fig. 2 | Context of the Mata Menge hominin fossils. a** Digital Elevation Map (DEM) of Flores showing the location of the So'a Basin and the cave Liang Bua. **b** DEM of the So'a Basin showing the location of the Mata Menge excavations. **c** Photo of the west baulk of the southern excavation sector (Sector 32C) in the upper fossil-bearing interval at Mata Menge. Layer III is a reddish sandy paleosol, overlain with an erosional contact by a sandy fluvial layer (Layer II). Layers II and III are covered by a series of clay rich ashy mudflow units (Layers Ia-f). Deciduous canine SOA-MM10 was recovered at 5 cm below the top of Layer II (indicated with the blue dashed rectangle; the blue rectangle is enlarged in e). Also note the large Stegodon bones resting on top of Layer II and covered by the mudflow units. **d** Photo of the northwest corner of the excavation in sector 32 A, taken on 7 November 2013, four weeks after the retrieval of the hominin humerus fragment SOA-MM9. The fossil was excavated from the top of Layer II, with the approximate position indicated with the dashed blue oval. **e** Detail of the contact between Layer I and Layer II at the spot of the deciduous canine SOA-MM10. **f** SOA-MM10 still partly embedded in the sandstone of Layer II. **g** Mata Menge excavation grid (UTM Zone 51 L) showing the 1×1 m quadrants excavated towards the end of the 2016 field season in gray. The positions of the hominin fossils described in this paper are indicated with green dots, those described previously[9] with gray dots. Light shading represents the step trench excavated in 2010, which first revealed the presence of the Mata Menge upper fossil-bearing interval bone bed. **h** SOA-MM11 surrounded by its sandstone matrix when excavated in 2016. The maps (**a** and **b**) created with GeoMapApp (www.geomapapp.org) / CC BY / CC BY (Ref. 67)".

lateral surface but on the posterolateral aspect, seen as a flattened area in CT slice nos. 1900 and 2000 ('RS' in Fig. 5); 2) in lateral view, the anterior margin exhibits a slight concavity around no. 1900 ('AM' in Fig. 5, see the surface rendered image on the left side), indicating that this part, which is ~13 mm below the preserved proximo-posterior end, leads to the deltoid tuberosity proximally. The distal margin of the deltoid tuberosity is situated on the antero-lateral surface at the 48.4% level in our modern human sample mean (N = 366, range: 43−53%), 51% in LB1[24], and 48% in KNM-WT 15000[25];

3) NF is present 21 mm distal to the preserved proximo-posterior end of SOA-MM9. The projected distance along the shaft from the 50% level and the lower margin of the NF is 23 mm in LB1, 1 mm in KNM-WT15000, 15 mm in MH2 (*Au. sediba*)[26], and 21.2 mm in our modern human sample mean (N = 366, SD = 10.0 mm, range = −2 to 59 mm). Each of the above three characters shows substantial variation, but their simultaneous expression at the preserved proximal shaft strongly suggests that SOA-MM9's proximo-dorsal end was very close to the original 50% level. This positioning is consistent

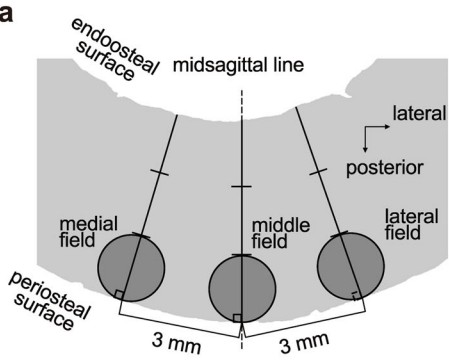

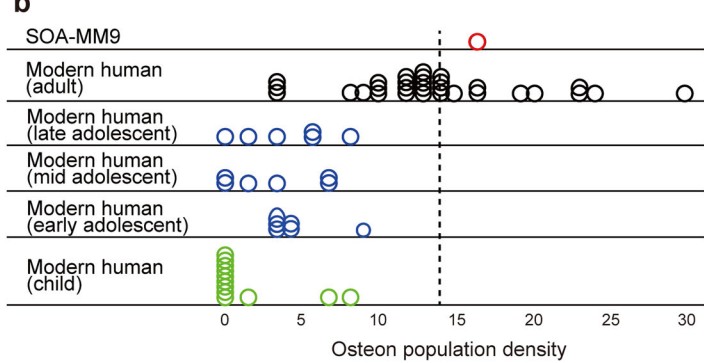

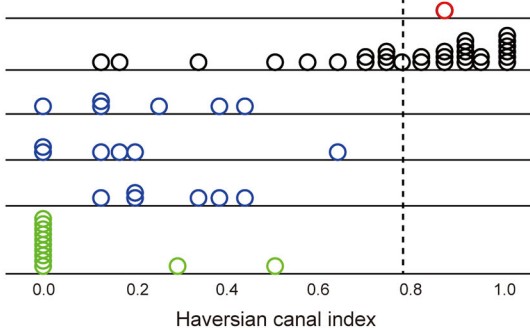

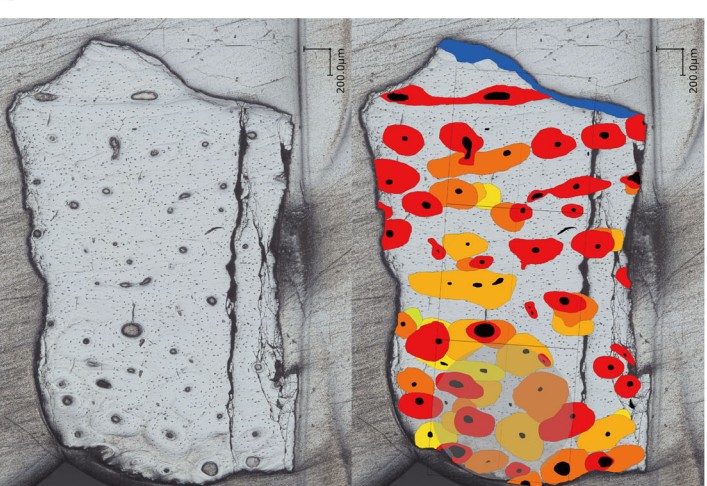

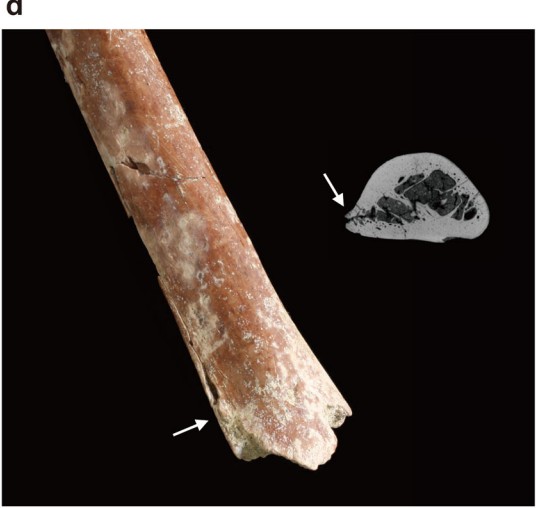

**Fig. 3 | Evidence for adulthood of the SOA-MM9 humerus. a** Three sites in a cortical section of the mid-posterior humeral shaft used for the histomorphological analyzes. 'Middle field' is a circle drawn in the outer one-third of the midsagittal line, and the 'lateral field' and 'medial field' are corresponding circles 3 mm apart from there. **b** Age-related histomorphometric values for SOA-MM9 and modern humans. The data from the three sites are plotted for all the modern human specimens, whereas the data for SOA-MM9 is from the lateral field only. The dotted lines are the means of the modern human adult subsample. See Supplementary Data 1 for the original data. **c** Cortical section of the posterior midshaft of SOA-MM9 observed by an ultra-high-definition microscope (VHX-7000, Keyence). The original image (left) and the same image colored for intact secondary osteons (red), fragmentary osteons (orange and yellow), and the intact inner surface (blue) (right). Note the dominance of secondary osteons around the exterior surface (downside of the images). Only one section (shown here) was produced to minimize the damage to the original specimen. **d** Lateral supracondylar ridge of SOA-MM9 and its CT section (the arrows). Note the weak but distinct development of the ridge as a slight eversion.

with other shaft morphologies exhibited by SOA-MM9 (Supplementary Note 3, Fig. 5, Supplementary Figs. 1 and 2, and Supplementary Data 4).

Next, distally, the CT section no. 250 of SOA-MM9 (Fig. 5), which is sliced at the hOF point, corresponds to the 12.5–14% level (or 11.5–15% more broadly) of maximum length (as explained below). Because of the observed allometric relationships that shorter modern human

humeri tend to have relatively large distal epiphyses (Supplementary Note 3), we referred to the following two samples to draw upon the above figures. One is the short-statured prehistoric Holocene population from Tanegashima Island, Japan (N = 13, maximum humeral length: 245–292 mm), and the other is a subset of short humeri from the prehistoric Jomon population from Japan (N = 10, maximum humeral length: 240–250 mm). The means of the hOF levels in these

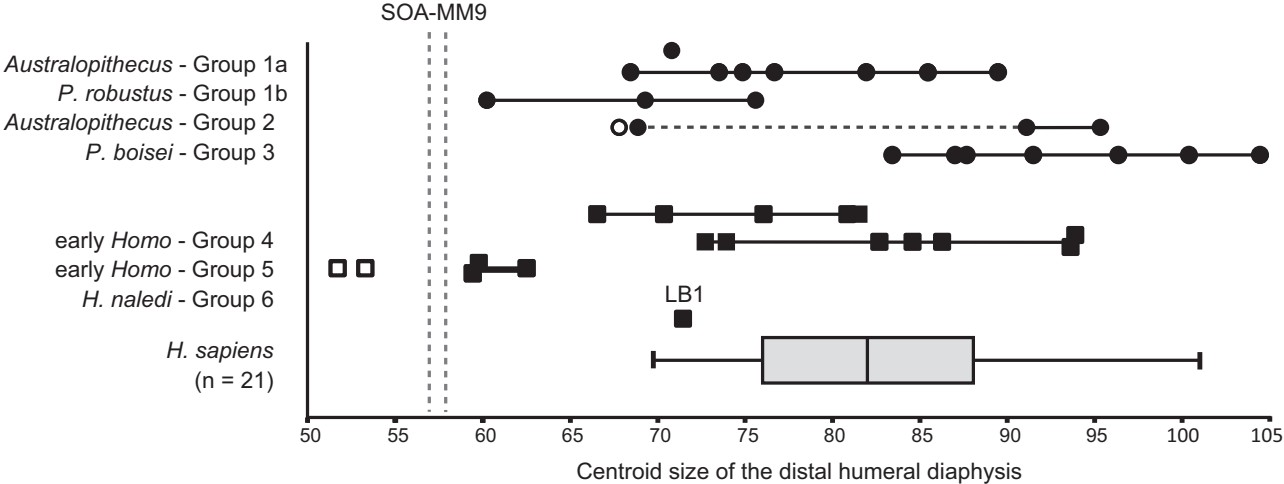

**Fig. 4 | Centroid size of the distal humeral transverse (~19%) section.** Symbols: circle = australopith; square = *Homo*; black = adult (possibly including some late adolescents); white = subadult. The specimens in each morphological group are listed in Supplementary Data 3 along with their size values. A standard box plot (the median at horizontal line and the 'whiskers' representing minimum/maximum values) is shown for a sample (N = 21) of modern humans. Transverse dashed line for Group 2 connects specimens belonging to *Au. sediba* and *Au. sp. indet*. Vertical dashed lines represent the smallest (proximal section) and largest (distal section) values obtained for SOA-MM9, which is smaller than all adult Plio-Pleistocene fossil hominin humeri (including LB1) and similar in size to specimens of *H. naledi*. Note that fully adult status (proximal epiphyseal fusion or developed muscle markings on the shaft) cannot be confirmed for some specimens such as the smallest individuals of Group 1b (SKX 10924) and Group 4 (SK 2598 and SK 24600). See Supplementary Data 3 for other notes.

samples were 12.5% and 14%, respectively, with their ranges 11.5 − 13.5% and 13 − 15%, respectively. The equivalent values in short hominin fossil humeri are 13% in both LB1 and A.L. 288-1.

Based on the above evaluations of SOA-MM9 humeral shaft preservation (proximal, 50% level; distal, 12.5 − 14% or 11.5 − 15% level), the original maximum humeral length of SOA-MM9 is estimated to be 211−220 mm or 206−226 mm, respectively. Alternatively, if we apply the mean ratios between the NF-hOF length and the maximum humeral length in our modern human male and female samples (0.29 to 0.30, Supplementary Data 2), the estimated maximum length of SOA-MM9 is 194−200 mm, but we surmise that this is less reliable given the weak correlation between the two measurements ($r = 0.37$).

**Comparative humeral morphology**

SOA-MM9 lacks characteristic features of *Australopithecus* distal humeri, such as a prominent flange-like lateral supracondylar ridge, a projecting medial supracondylar crest, and marked curvature in the sagittal plane, although expression of these traits tends to be weak in comparatively gracile specimens of this genus[26–29]. With respect to cross-sectional shape (distal shaft 19% level), SOA-MM9 is similar to small-bodied *Homo* (*H. naledi* and *H. floresiensis*) in having a medio-laterally narrow profile that is unusual among the comparative groups (Fig. 6). It is different, however, from small-bodied *Australopithecus* individuals such as A.L. 288-1, whose humerus is only slightly longer than estimated for SOA-MM9 (Fig. 6). Procrustes distances support the cross-section shape variation summarized by the PCA results. Whereas the distances of SOA-MM9 to group mean shapes (Supplementary Fig. 4a) or individual specimens (Supplementary Fig. 4b) of *H. naledi* and *H. floresiensis* do not exceed the degree of within-species variation represented by our modern human sample, the same distances to the other fossil taxa are much greater.

**Maxillary deciduous canine (d$^c$: SOA-MM10)**

This right tooth preserves a complete crown and a broken, short segment of the root (Fig. 1j). The crown is extremely small, situated well below the reported range of *H. sapiens* (Fig. 7a, and Supplementary Table 1), in a similar way to the previously reported Mata Menge d$_c$s (Fig. 7b). The specimen has a primitive, relatively low distal shoulder that resembles *Australopithecus* and Sangiran *H. erectus* homologs,

although PCAs based on four or five linear measurements indicate that this morphology is marginally within the large variation seen in *H. sapiens* (Supplementary Fig. 5). Occlusal wear exposes a small dentine patch on the cusp tip and a thin line of dentine on the elongated distal incisal margin. The latter suggests the presence of a primitive, tall mesial cusp configuration on its occluding deciduous first molar (dm$_1$), as known for an early Javanese *H. erectus* dm$_1$ from Sangiran, S7−67[30].

**Mandibular third molar (M$_3$: SOA-MM11)**

The preserved left crown has reduced distal cusp areas, distally protruding hypoconulid, and no distal interproximal facet (Fig. 1k). Wear has flattened much of the occlusal surface, except for the metaconid that remains relatively high. Crown diameters are comparable to those of the smaller Liang Bua individual (LB6/1) and are marginally within the large variation exhibited by our global *H. sapiens* sample (Fig. 7c). It has five principal cusps arranged in a '+' pattern and is different from the mandibular third molars of Liang Bua *H. floresiensis* that exhibit a derived, four-cusped morphology (LB1, LB6/1)[31,32]. Occlusal crown contour examined by normalized Elliptic Fourier Analysis shows that it clusters firmly with Sangiran *H. erectus* and marginally with *H. ergaster*, having a mesiodistally short crown. It is outside the range of variation exhibited by *H. habilis sensu lato*, which is primarily characterized by a mesiodistally elongated and distally tapered crown (Fig. 7d, Fig. 8, Supplementary Note 4, Supplementary Fig. 6) with a tendency for better developed hypoconulids and accessory cusps[33–35].

**Comparison of size with the Liang Bua *H. floresiensis***

In all available measurements of the mandibular body, postcanine teeth (P$^3$, M$_{1/2}$ and M$_3$) and distal humerus, the Mata Menge fossils reported here or previously[9] are smaller than the Liang Bua *H. floresiensis* remains (LB1 and LB6/1) by 1−21% (Table 2). Stature estimates, based on humeral length of SOA-MM9 (211−220 mm) and LB1 (243 mm), are 103−108 cm and 121 cm, respectively, using the human pygmy model[36]; or 93−96 cm and 102 cm, respectively, using the ape model[37].

## Discussion

All the ten hominin remains so far discovered from Mata Menge were excavated from a narrow area (about 7 m × 20 m) within the upper part

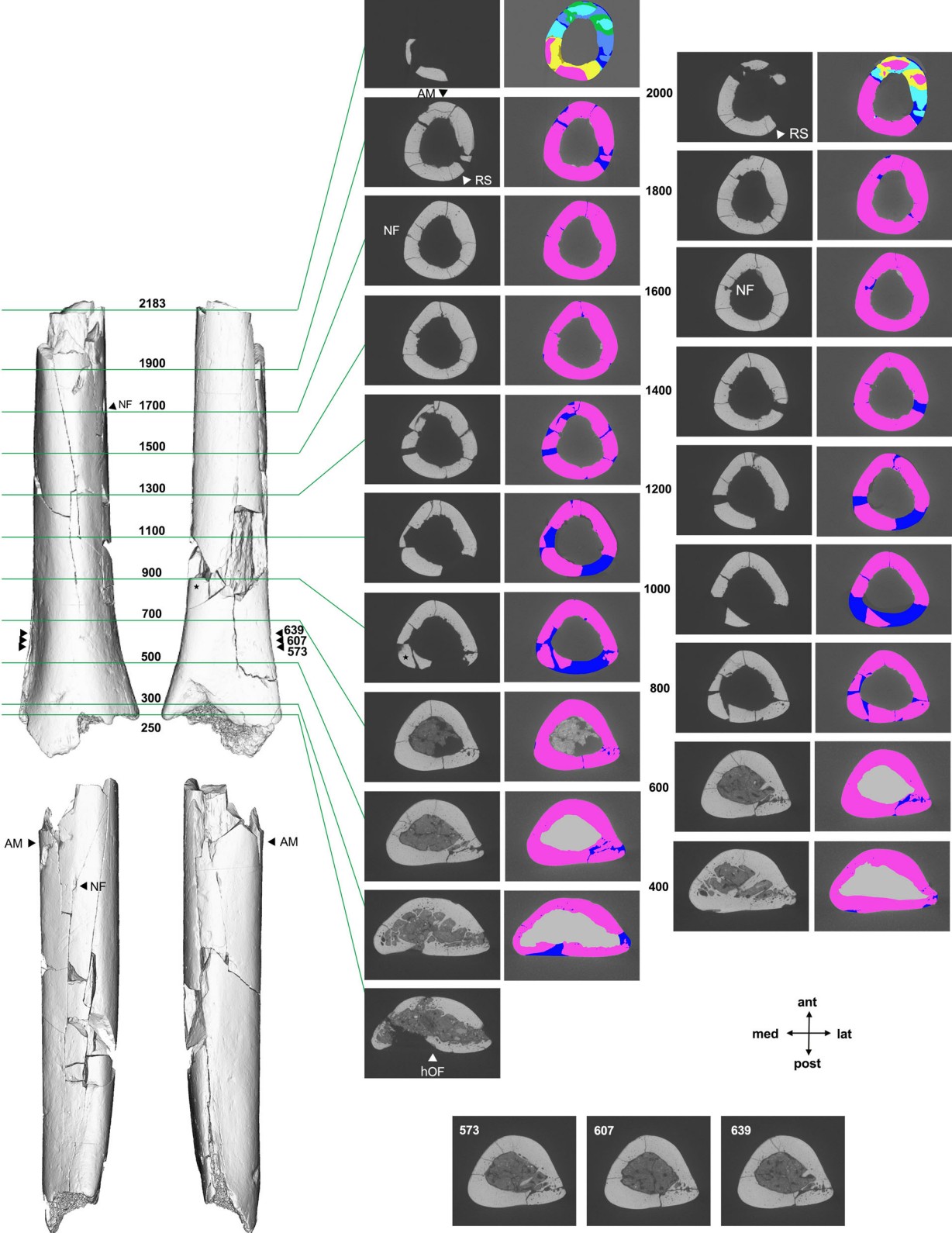

**Fig. 5 | CT-based images of SOA-MM9.** Left: Surface rendered images. Clockwise from top left: anterior, posterior, right lateral and left lateral views. Right: CT sections and reconstructed cortical bones at the slice level indicated by the numerals (250 – 2183). The slice nos. 607, 573 and 639 are estimated 19% levels (the best estimate and probable range: see "Methods"). The slice thickness of these CT sections is 0.04 mm, so that the difference of 100 corresponds to 4 mm. Cortical bone reconstruction was made with reference to the intact bones in the same or adjacent slices. The preserved cortical bones are in pink, the extrapolated portions are in blue, and the regions 'transplanted and trimmed' from nearby slices are in other colors. No. 900 was reconstructed after a minor positional correction of the small bone indicated by the star, which is slightly dislocated inward in the current reconstruction.

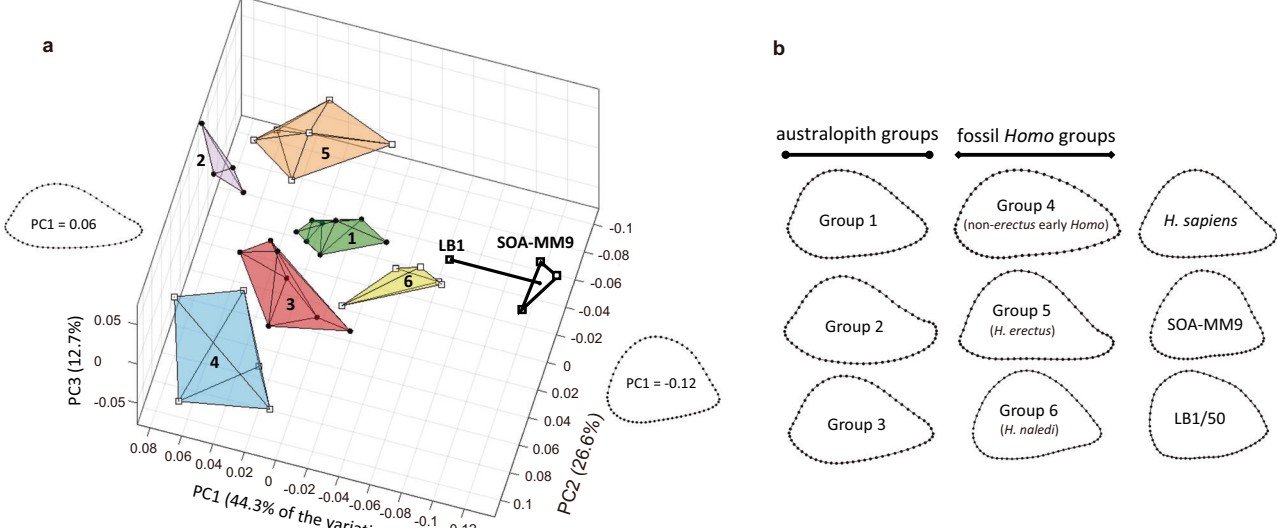

**Fig. 6 | Geometric morphometric analysis of humeri. a** Principal component analysis (PCA) of distal diaphyseal shape among 40 fossil hominin humeri. Convex hulls define the fossil groups (Groups 1-3 = australopith, Group 4 = *H. habilis*, Group 5 = *H. erectus s.l.*, Group 6 = *H. naledi*: see Supplementary Data 3 for more details) and the dotted line connects LB1 to the average shape of the three sampled sections of SOA-MM9. The two diaphyseal outlines depict shape variation exclusively along PC1. SOA-MM9 is extreme along PC1 and is most similar in overall shape (based on Procrustes distance) to LB1 and to specimens of *H. naledi* (Group 6). See Supplementary Fig. 3 for a two-dimensional presentation of this result. **b** Cross-sectional outlines of the distal humeral diaphysis of SOA-MM9 and LB1 in comparison to group averages for modern humans and the fossil hominin morphological groups. Shape configurations are shown scaled by anteroposterior width (with anterior towards the top and lateral to the right).

of Layer II (Fig. 2). We previously reported that one mandible fragment (SOA-MM4) and six isolated teeth (SOA-MM1, 2, 5, 6, 7 and 8) from this collection represent at least one adult and two children[9] (Table 1). The limited wear on the mesial incisal margin of the new right d$^c$ (SOA-MM10) does not match the extensive wear on the distal incisal margin of the previously reported right d$_c$ (SOA-MM8), but the degree of wear does not preclude the possibility that SOA-MM10 and SOA-MM7 (left d$_c$) are from the same child. While SOA-MM7 was recovered from sieving, this specimen and SOA-MM10 were found within 6 m horizontal distance of each other. The new permanent molar (SOA-MM11), a moderately worn left M$_3$, is obviously a different individual from SOA-MM1, a lightly worn left M$_1$ (or M$_2$) belonging to an adolescent (or a young adult if this was a M$_2$). Therefore, the current Mata Menge hominin assemblage includes at least four individuals including one adult, one adolescent/young adult, and two children (Table 1).

The observation that all four (or more) individuals are extremely diminutive supports the argument that small body size was not an idiosyncratic (individual) character but a population feature of the early Middle Pleistocene hominins of Flores. The markedly small deciduous teeth from at least two individuals, which are almost outside the large variation range of modern humans (Fig. 7), also indicate that the Mata Menge hominins had diminutive dental size at birth. Additionally, the strikingly small adult humerus (SOA-MM9) reported here demonstrates that this character was not limited to the dentognathic elements but also extended to upper arm size. On this note, it is worth highlighting that the two or more Mata Menge adult/adolescent individuals are consistently smaller than the two adults of Liang Bua *H. floresiensis* (Table 1). This strongly suggests that by ~0.7 Ma, hominins on Flores were already as small as, or perhaps slightly smaller than, the Late Pleistocene *H. floresiensis* (Supplementary Note 5).

Based on the previously recovered dentognathic sample, it was suggested that the Mata Menge fossils could be reasonably assigned to *H. floresiensis*[9]. Now that a new arm bone and additional dental remains belonging to this assemblage display strong affinities with the Liang Bua remains, we can more confidently classify these early Middle Pleistocene hominins into *H. floresiensis*. Notable minor differences between the two widely separated chronological forms include the lack of molar morphological specializations (see below) and possibly the smaller body and dental sizes in the earlier Mata Menge hominin.

This study also contributes to the debate over the origin and evolution of *H. floresiensis*. It was previously reported that the Mata Menge hominins now assigned to *H. floresiensis* were more similar to early Javanese *H. erectus* than to *Australopithecus* and *H. habilis sensu lato* in mandibular body form and M$_1$ (or M$_2$) shape[9], a finding that runs contrary to hypotheses that assume a direct evolutionary link between *H. floresiensis* and pre-*H. erectus* hominins such as *H. habilis*[12–14]. The present study indicates that the shape similarity between the Mata Menge fossils and early Javanese *H. erectus* also applies to M$_3$, and that the Mata Menge molars lack the unique specialization seen in the Liang Bua *H. floresiensis* homologies (i.e., four-cusped, mesiodistally shortened and somewhat distorted molar crowns[9,32,37]). Therefore, archaic *H. floresiensis* at Mata Menge probably represents the dwarfed lineage of early Javanese *H. erectus* at a stage prior to unique molar specializations. Alternatively, if *H. habilis s.l.* was ancestral to Mata Menge/ Liang Bua *H. floresiensis*, the latter would need to have experienced substantial molar size reduction of ~65–60% in mesiodistal and buccolingual crown diameters (from *H. habilis* means), and this accompanied by form changes comparable to the early Javanese *H. erectus* condition. Because no such allometric relationships are evident between molar crown size and form within *H. habilis* (Supplementary Note 6), the hypothesis that *H. floresiensis* is a direct lineal descendant of *H. habilis s.l.* is not supported. In contrast, molar size reduced from the Lower to Upper Sangiran dental assemblages (Fig. 7c) without significant form changes (Fig. 7d), confirming that such local evolution could occur. Additionally, although the humeral shaft morphology of SOA-MM9 does not indicate an affinity with either *H. erectus* or *H. habilis*, its cross-sectional shape is most similar to that of dwarfed taxa of *Homo* (*H. floresiensis* and *H. naledi*) and unlike that of small-bodied *Australopithecus* individuals.

Coupled with the recently revised arrival date for *H. erectus* on Java ( ~1.1 Ma, or at most younger than 1.3–1.5 Ma)[11] and hominins on

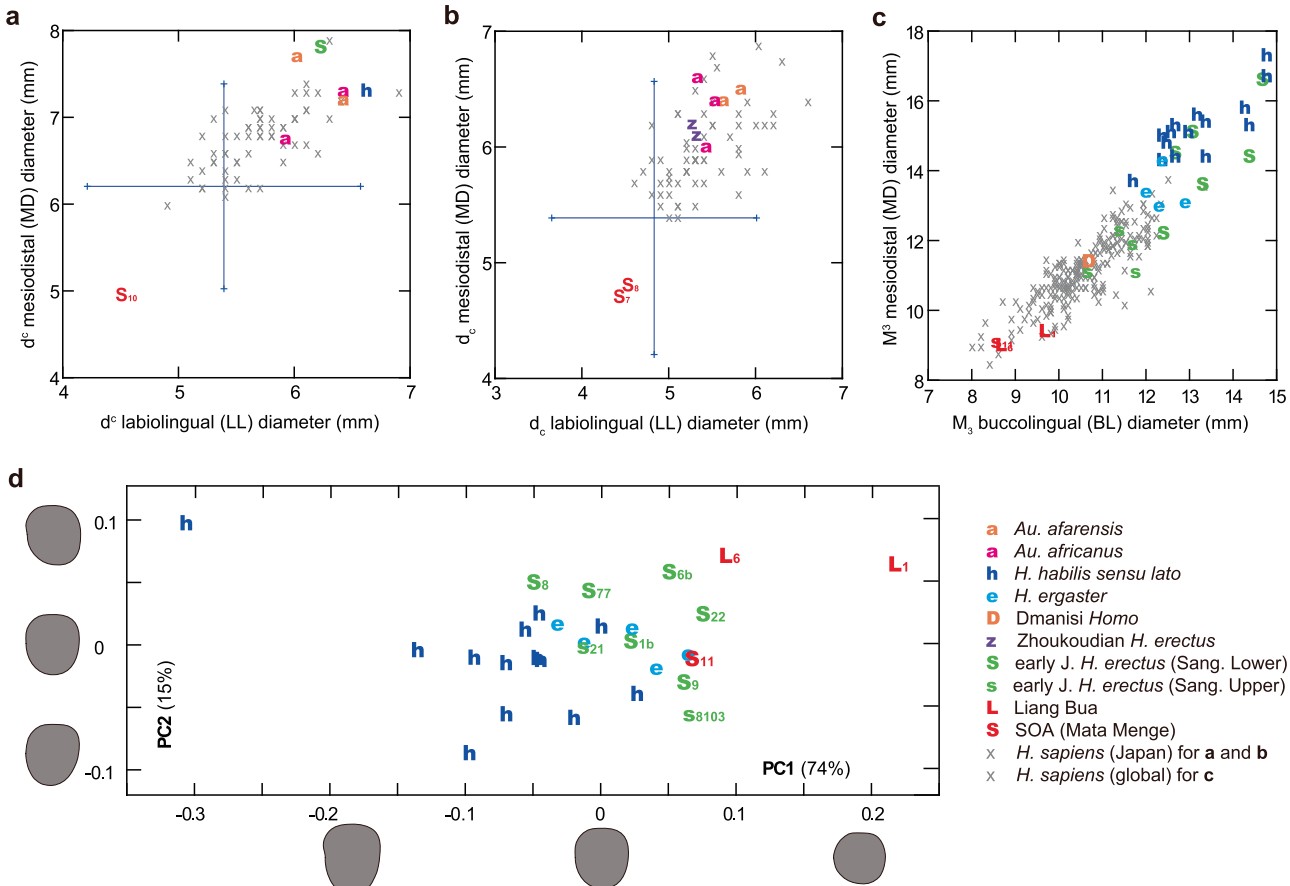

**Fig. 7 | Metric comparisons of dental remains.** Horizontal crown dimensions of maxillary deciduous canines (dᶜ: **a**) and mandibular third molars (M₃: **c**), as well as the previously reported mandibular deciduous canines (d_c: **b**). The large crosses in a and b indicate 2 SD ranges for the smallest-toothed modern population as for dᶜ and d_c (India[66]). **d** Plots of PC scores derived from the normalized Elliptic Fourier Analysis (EFA) of M₃ crown contour. Shape differences along PC axes are shown on left teeth for two standard deviations from the origin. Symbols: 'a' (orange) = *Au.* *afarensis*, 'a' (magenta) = *Au. africanus*, 'h' (blue) = *H. habilis sensu lato*, 'D' (orange) = Dmanisi *Homo*, 'z' (violet) = Zhoukoudian *H. erectus*, 'S' (green) = early *H. erectus* (Sangiran Lower), 's' (green) = early *H. erectus* (Sangiran Upper), 'L' (red) = Liang Bua, 'S' (red) = Soa (Mata Menge). 'x' = *H. sapiens* (Japan) for (**a**) and (**b**) *H. sapiens* (Japan) for (**c**). Specimen numbers are indicated for selected samples in each symbol. Source data are provided as a Source Data file.

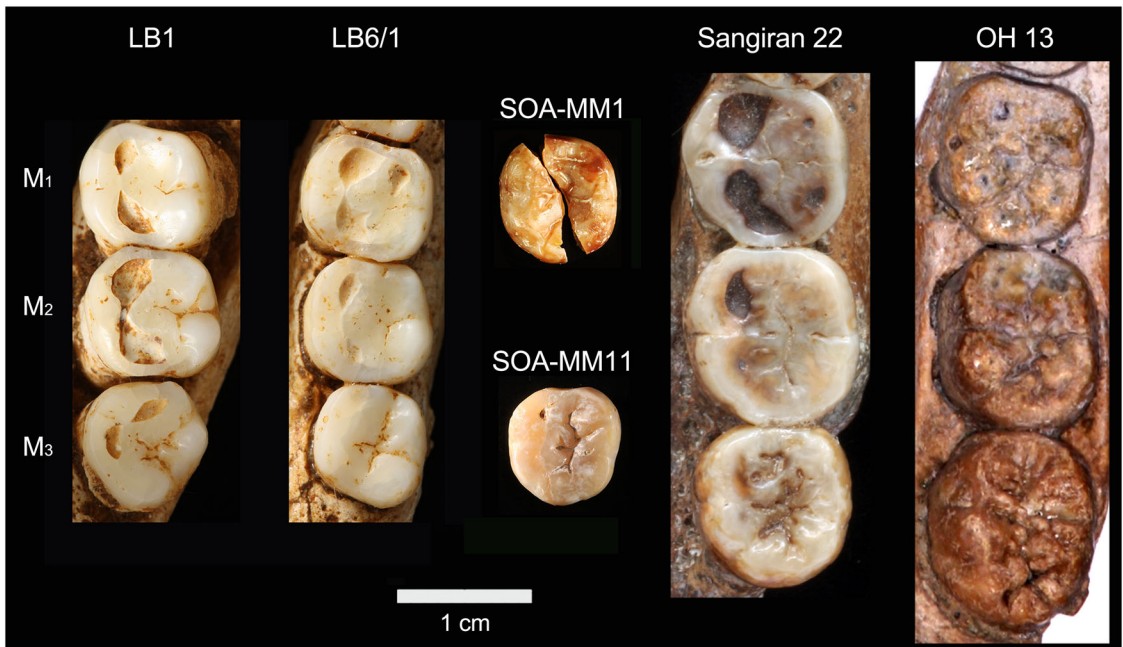

**Fig. 8 | Mandibular molars of selected fossil *Homo* individuals.** LB1 and LB6/1: Liang Bua *H. floresiensis*, Sangiran 22: early Javanese *H. erectus*, OH13: *H. habilis*.

**Table 2 | Relative size of the Mata Menge specimens as compared to Liang Bua _H. floresiensis_**

| | Mata Menge (MM) | | LB1 | | LB6 | |
|---|---|---|---|---|---|---|
| | | | % | mm | % | mm |
| **Mandible** | | | | | | |
| Mandibular corpus height at M$_{2/3}$ | 18.5 | SOA-MM4 | −21 | 23.4 | −20 | 23.1 |
| Mandibular corpus breadth at M$_{2/3}$ | 13 | SOA-MM4 | −18 | 15.8 | −17 | 15.7 |
| **Teeth** | | | | | | |
| Sqaure root of calcurated area (P$^3$) | 8.0 | SOA-MM5 | −1 | 8.1 | | |
| Sqaure root of measured area (M$_1$) | 8.4 | SOA-MM1 (M$_{1/2}$) | −8 | 9.1 | −3 | 8.7 |
| Sqaure root of measured area (M$_2$) | 8.4 | SOA-MM1 (M$_{1/2}$) | −8 | 9.1 | −4 | 8.7 |
| Sqaure root of measured area (M$_3$) | 7.9 | SOA-MM11 | −6 | 8.4 | −1 | 8.0 |
| **Humerus** | | | | | | |
| NF-hOF length* | 58 | SOA-MM9 | −9 | 64.0 | | |
| Minimum circumference of the shaft | 46 | SOA-MM9 | −16 | 55.0 | | |

Relative size (%) and individual measurements (mm). The formers were calculated as (LB-MM)/LB*100.

*Projected length between the NF and hOF points (see "Methods").

Flores (1.0–1.27 Ma)[6], as well as the reported craniometric and odontometric analyzes which almost unanimously support strong affinities of _H. floresiensis_ with _H. erectus_ (particularly early _H. erectus_ from Java)[37–42], the following evolutionary model emerges. The earliest Flores hominins appeared on this Wallacean island ~1.0–1.27 Ma, probably unintentionally (i.e., through accidental 'rafting', perhaps on tsunami debris), and possibly as part of the initial colonization of the Sunda Shelf region by early _H. erectus_. The Flores hominins experienced substantial body size reduction soon after this event (within ~300,000 years), despite the presence of large-bodied predators such as ~3 meter-long Komodo monitors and crocodiles from the earliest paleontological record ( ~1.4 Ma) onwards[6]. This implies that giant reptilians did not represent a serious predation threat for early _H. floresiensis_ or its progenitors. This early evolutionary event was followed by long-term stability in hominin body size, possibly also in cultural adaptations (e.g., stone technology[6–8]), and minor morphological specialization in the dentition. How the small brain size reported for the ~60,000 years old LB1[1,43] evolved still remains unknown. At present, however, the available fossil data imply that small body size had been a functional adaptation for these insular hominins during and slightly beyond the Middle Pleistocene and indeed potentially up until the arrival of _H. sapiens_ on Flores around 50,000 years ago; an event that, we suspect, precipitated the demise of _H. floresiensis_[10].

## Methods

Permission to undertake excavations at Mata Menge was granted by the Indonesian State Ministry of Research and Technology (RISTEK permits 300/SIP/FRP/SM/VIII/2013 and 2183/FRP/SM/X/2015), the provincial government of East Nusa Tenggara in Kupang, and the Ngada District Administration. CT scan of SOA-MM9 was conducted in Tokyo with a permission issued in 2015 from the Geological Agency, Bandung.

**CT scan and measurements**
A micro-CT scan of SOA-MM9 was taken by using TXS320-ACTIS (Tesco Co.) at the National Museum of Nature and Science, Tokyo, with the following scanning parameters: 205 kV and 0.2 mA with a 0.5 mm thick copper plate prefilter, a 1024 × 1024 matrix, 0.04 mm pixel size and 0.04 mm slice interval (0.043 mm slice thickness). Micro-CT scans (voxel size = 0.156 mm) of 88 adult prehistoric Japanese (Holocene hunter-gatherer-fishers from the Jomon period) were also obtained for the length estimation of SOA-MM9 (see below). Linear measurements were taken using a spreading digital caliper (to the nearest 0.1 mm), an osteometric board (to the nearest 1.0 or 0.5 mm) and measuring tape (to the nearest 0.5 mm). Cross sectional properties were calculated based on CT scans at 15–50% vertical level of the shaft, using the software CT-Rugle (ver. 1.2, Medic Engineering Inc., Japan) and ImageJ (ver. 1.53f51, National Institutes of Health, USA).

**Humeral analyzes**
**Comparative samples.** To characterize its humeral morphology as a specialized insular hominin group, SOA-MM9 was compared with a wide variety of Pliocene and Pleistocene Afro-Asian hominin humeri (_Ardipithecus_, _Australopithecus_, _Paranthropus_, African early _Homo_, Dmanisi _Homo_, _H. erectus/ergaster_ and _H. naledi_), as well as _H. floresiensis_ from Liang Bua (LB1) and a series of modern human samples including some short-statured populations. The individual specimens included for linear metric comparison and geometric morphometric analysis are shown in Supplementary Data 2 and 3, respectively. The modern human samples used for the linear metric analysis are in Supplementary Data 2, while the modern human sample for the geometric morphometric analysis (Supplementary Data 3) is a mixed-sex sample of adults collected (by J.M. Plavcan) at the Smithsonian National Museum of Natural History (Washington, DC)[44]. As for the geometric morphometric analysis, the fossil sample was divided into six morphological groups, five of which have been established by previous studies[29,44–46], while a sixth group consists of five specimens attributed to _Homo naledi_. We also collected outline data from a scan of the humerus of the LB1 skeleton (i.e., LB1/50) attributed to _H. floresiensis_[1]. The adult/subadult status for each specimen was determined by the epiphysial fusion of the proximal and/or distal ends, or other information if available (e.g., dental development). The linear metric comparisons focus on the adult samples, while the geometric morphometric analysis contains some subadult specimens, as noted in Fig. 4 and Supplementary Data 3. See Supplementary Note 7 for additional information about the _H. naledi_ sample.

**Key landmarks.** SOA-MM9 lacks most of the widely used osteometric landmarks, but the following points are usable:

Nutrient foramen (NF): the distal margin of the nutrient foramen on the midshaft.

hOF point: the proximal margin of the hollow leading to the olecranon fossa.

**Developmental age.** Human bones undergo substantial histomorphic changes during development and much of adulthood. In a limb bone shaft, periosteal cortical bone growth occurs as deposition of circumferential lamellar bone and primary osteons with non-Haversian canals. The proportion of these primary structures decreases as secondary osteons (Haversian canal surrounded by concentric rings of lamellar bone) appear and increase through bone remodeling. In late adulthood, the cortical bone is dominated by secondary osteons[17,47]. This process is numerically demonstrated by the count or density of the elements of secondary osteon (intact osteons, fragmentary osteons, Haversian canals, resorption bay, etc.)[20,48–50]. However, because the rate of such histomorphic change varies considerably depending on the locus in a bone, regional differences must be considered in histomorphological age estimation[21,51]. Because no histomorphometric data was available in the literature for human humeral midshaft, we collected our

referential data using modern Japanese humeri. Our sample, which consists of 10 adults, 6 adolescents, and 4 child individuals, was unearthed from cemeteries of the Edo period (17th-19th centuries A.D.) in Tokyo City and is stored at the National Museum of Nature and Science, Japan. Based on the standard ossification procedure[52], we categorized those humeri with completely fused epiphyses to 'adult,' those with unfused proximal epiphysis and completely fused distal epiphysis to 'adolescent,' and those with separate epiphyses as 'child.' The adolescent category was further subdivided into 'early adolescent' (fusion at distal epiphysis only, N = 2), 'mid adolescent' (fusion at distal epiphysis and medial epicondyle, N = 2), and 'late adolescent' (proximal epiphysis partially fused, N = 2). We (J.S.) cut out a small piece of the bone from the mid-posterior shaft of the SOA-MM9 humerus, to prepare a sectional sample ('HS' in Fig. 1c). The location of this section is 6.5 mm distal to the preserved proximal edge, and 14.5 mm proximal to the distal margin of the nutrient foramen, and is assumed to be slightly distal to the missing deltoid tuberosity. The obtained section covers a full thickness from its outer (periosteal) to inner (intrathecal) surfaces. After embedding in resin, we prepared a polished surface to observe with an ultra-high-definition microscope (VHX-7000, Keyence). From each of the modern human humeri, we prepared a cortical section at the posterior surface 10 mm distal to the lower end of the deltoid tuberosity. First, we cut out a small piece of the bone using a diamond cutter. After embedding in resin, a transverse section of 70 μm-thickness was made by a microtome (SP-1600, Leica) to observe under an ordinary light microscope (ImagerA1, Leica). We focused on the outer one-third of the cortical bone, because the periosteal region is an active bone growth field and is useful for histomorphometric growth studies[20]. To allow for regional variation mentioned above, we examined three adjacent loci: one on the midsagittal line (middle field), and the others on either side of it (lateral and medial fields) as illustrated in Fig. 3. The section prepared for SOA-MM9 corresponds to the lateral field. In each observation field, we counted the numbers of intact secondary osteons (N.On), osteon fragments (N.Fr), resorption bays (N.Re), Haversian canals of the secondary osteons/osteon fragments (N.Ca), and non-Haversian canals (N.nCa), as defined elsewhere[20,53]. An 'osteon fragment' is a secondary osteon eroded by later-formed osteons. A structure straddling the border of the observation field was counted only if more than half of it was inside. We use the following three size-free parameters as measures of cortical bone growth and pathology.

1) Osteon Population Density (OPD): This widely used index, calculated here as (N.On+N.Fr+N.Re) per area (mm2)[21], monitors the increase of secondary made structures. A greater OPD value reflects advanced growth stage.

2) Haversian Canal Index (HCI): This is a ratio of the secondary made canals. It is calculated as (N.Ca/(N.Ca+N.nCa)), and increases from 0 to 1 with bone growth.

3) Relative Cortical Bone thickness (rCBt): We define this index as the mean cortical bone thickness divided by the minimum circumference of each humeral shaft. The former is the average of the three cortical bone thicknesses at the medial, middle and lateral fields in Fig. 3, which we measured using a public-domain ImageJ (U.S. National Institutes of Health, available at https://imagej.nih.gov/ij/).

**Length estimation.** The extant and fossil hominin humeri exhibit a uniform pattern of transition in cross-sectional shape (i.e., flatness, angle of the long axis, ratio between the cortical and total areas, and other features) from the mid- to the distal shaft levels (Supplementary Figs. 1 and 2). We (S.M. and Y.K.) refer to this information to reconstruct the original maximum humeral length (Martin no. 1a) of SOA-MM9.

**Cross-sectional properties of the shaft.** We (S.M.) used CT-Rugle 1.2 (Medic Engineering Co.) to calculate the cross-sectional properties of the humeral shaft.

**Cross-sectional geometry of the shaft.** Previous studies of fossil hominin humeri have demonstrated the taxonomic utility of the cross-sectional shape of the distal diaphysis sampled at ~19% of total (biomechanical) humerus length from the distal end[29,44,46,54–56]. The ~19% level of SOA-MM9 was located by Y.K. based on our estimate of its maximum length (211–220 mm), which was converted to the biomechanical humeral length using the ratio between the two (the former is 1.08% longer on average in our mixed-sex, prehistoric modern human (Jomon) sample: N = 88). The 19% level thus located is within the CT slice nos. 573–639. Therefore, we chose three slices, nos. 573, 607 (best estimate), and 639 for the present analysis. Two-dimensional coordinates were collected by M.L. from all three sections of SOA-M9 following the procedure described previously[44] (i.e., two Type 2 landmarks on the medial and lateral extremes of the specimen and 58 sliding semilandmarks on the anterior and posterior surfaces). Raw landmark configurations were superimposed into the same shape space using orthogonal least-squares generalized Procrustes (GPA) superimposition[57], GPA was performed using tpsRelw software[58] and semilandmarks were allowed to slide along the diaphyseal outline using the criterion of minimized bending energy[59]. Subsequent to GPA, morphometric relationships were assessed with the use of Procrustes distances (Dp) as a measure of shape dissimilarity[60] and principal component analysis (PCA) as a means of visual summary (via ordination) of shape variation among the individual specimens.

### Dental analysis

SOA-MM10 (maxillary deciduous canine: d$^c$) was compared with the available sample of fossil hominins (*Australopithecus* and Early Pleistocene *Homo*), as well as a sample of *H. sapiens* (Supplementary Table 2) by Y.K. Unfortunately, there is no deciduous teeth in the existing *H. floresiensis* assemblage from Liang Bua. SOA-MM11 (mandibular third molar: M$_3$) was compared with Liang Bua *H. floresiensis* and its claimed two major ancestral candidates, *H. habilis* and early Javanese *H. erectus* (Supplementary Table 2). The early Javanese *H. erectus* dental sample examined in this report is from Sangiran, Central Java. We divided this sample into two chronological subsamples, Sangiran Lower and Sangiran Upper, following the previous report that demonstrated significant morphological differences in tooth size, mandible features, cranial capacity, etc.[61,62]. Linear measurements were taken based on the original specimens or high-quality casts by Y.K. using a digital caliper (Mitsutoyo Inc.) or otherwise collected from the literature (Supplementary Table 2). Occlusal crown contours of SOA-MM11 was further analyzed by normalized elliptic Fourier analysis (size-standardized EFA), using the comparative samples shown in Supplementary Table 2 and based on the methods detailed elsewhere[37]. In brief, the occlusal contour of each specimen was obtained from a photograph of the original specimen or high-quality cast, in a way which minimizes the error derived from parallax effect and orientation of the tooth or scale. Images were uploaded into Canvas X software (ACD Systems) to extract the occlusal contour and, for a worn tooth, to reconstruct small parts of the crown lost by interproximal wear. Then, normalized elliptic Fourier analysis was conducted using the software SHAPE 1.3[63], after each crown contour was aligned along its mesiodistal axis. We did not assess sex for these materials because of the small sample and the reported low sexual dimorphism in modern human deciduous teeth[64].

### Reporting summary

Further information on research design is available in the Nature Portfolio Reporting Summary linked to this article.

## Data availability

All data generated or analyzed during this study are included in this published article (and its supplementary information files) or as a Source Data file. The Source Data file includes raw data used for Figs. 6 and 7, and Supplementary Figs. 2, 3, 4 and 6. The Mata Menge hominin fossils are housed at the Geological Museum, Bandung. The 3D data of SOA-MM9 humerus may be shared on request to Unggul P. Wibowo (unggul.pw@esdm.go.id). Source data are provided in this paper.

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

## Acknowledgements

The former head of the Geological Agency, Banding, Dr. Surono, is acknowledged for his support. Geodetic recordings of excavated finds were conducted by Y. Sopyan (2013), and E.E. Laksmana and A. Rahmadi (2014). We thank the people from Mengeruda and Piga villages for their participation in the excavations and their ongoing support. We also thank the following persons for their contributions in the field during the 2013–2016 Mata Menge excavations: T. Suryana, E. Sukandar, A. Gunawan, Widji, A.T. Hascaryo, E. Setiyabudi, A.M. Saiful, B. Burhan, P.D. Moi, B. Alloway, B. Pillans, M. Storey, D. Yurnaldi, M. Moore, T. Sutikna, H. Insani, M.R.Puspaningrum, I. Yoga, H.J.M. Meijer, S. Hayes, and F. Aziz. We thank Takao Sato and Takashi Sano for their assistance in conducting morphological analyses, and J.M. Plavcan, C. Ward, M. Domínguez-Rodrigo, F. Di Vincenzo, and W. Kimbel for 3D scans and/or casts of fossil humeri. We are indebted to the following institutions: Pusat Penelitian Arkeologi Nasional, Senckenberg Research Institute and Natural History Museum Frankfurt, American Museum of Natural History, Sapporo Medical University, Niigata University, National Museum of Nature and Science, Tokyo, The University of Tokyo, St. Marianna University School of Medicine, Tahara Municipal Museum, Kyoto University, Kyushu University, Sasebo City Museum Shimanose Art Center, Okinawa Prefectural Museum and Art Museum, Okinawa Prefectural Archeology Center. Aspects of this research were financially supported by Australian Research Council Discovery grant (DP1093342 to the late M.J.M. Morwood and A.B.), Australian Research Council Future Fellowship (FT100100384 to G.D.vd.B.), Center for Geological Survey Bandung, Indonesia, Geology Museum Bandung, Indonesia (to I.K., R.S., I.S. and U.P.W.), JSPS KAKENHI grant (22H00421 to Y.K. and 23K17521 to J.S.) and National Science Foundation (BCS-0647557 to M.L.).

## Author contributions

Y.K., G.D.v.d.B., and I.K. conceived and led the study. Excavations were led by I.K. and R.S., with assistance from I.S. U.P.W. A.B. and G.D.v.d.B. Morphological analyses were conducted by Y.K., S.M., J.S., and M.L. with assistance from G.S., R.T.K. and T.S. Y.K., S.M., J.S., and M.L. wrote the manuscript with editorial inputs from all co-authors.

## Competing interests

The authors declare no competing interests.
