## [Peer Review File · Nature Communications]

Early evolution of extremely small body size in *Homo floresiensis*Reviewers' Comments:

Reviewer #1:

Remarks to the Author:

Kaifu et al. reported three new hominin fossils recovered from SOA-MM and assigned them, along with previously found materials, to *Homo floresiensis*. The more primitive traits and Sangiran affinity directly support the hypothesis that these dwarfed hominins evolved from *H. erectus*, which arrived here in the early Middle Pleistocene, or perhaps even earlier, as hinted by the evidence of artifacts. Have the authors considered whether the dwarfing process was gradual or a punctuational event, or if it relates to some environmental change? Regarding the cross-sectional geometry, the authors would better provide the original dataset instead of just figures. This will facilitate future comparisons.

Reviewer #2:

Remarks to the Author:

In this paper, the authors present new finds from Mata Menge, Flores, Indonesia. These add to the previously described material assigned to *Homo floresiensis*, dating to around 700 ka ago, significantly earlier than the eponymous material from Liang Bua (about 60ka). The most important specimen is an adult humerus, which is smaller and more gracile than that of the holotype, LB1. In addition to the humerus, two teeth were also recovered, a maxillary deciduous canine and a lower third molar, both of these are very small. Based on these new discoveries, the authors propose that *floresiensis* was present on Flores since about 700,000 years ago and probably evolved from earlier Asian *Homo erectus*.

Congratulations to the authors on such a well executed and interesting study. These discoveries contribute significantly to our understanding of *Homo floresiensis*, and especially to how and when the remarkably small body size in this species evolved. The authors have provided detailed descriptions of the specimens and of their analyses, and in general, this paper clearly warrants publication due to the importance of the material described within, and the very detailed analyses. There are a few issues though that I think if dealt with would further strengthen the paper:

1. I found it rather confusing and hard to understand which comparative sample was used where in the humeral analyses. Even though SI Table 2 lists a very large number of humeri used in the comparative analyses, only a subset of the modern comparative sample was used in the distal humeral GMM analyses. It would be better to be more explicit about this, so the reader doesn't have to hunt for this information in several different places.
2. Linked to the previous, why was the modern comparative sample for the GMM analyses of the distal humerus so small ($n=21$, at least according to the Ext. data Fig 7 legend), when compared with the sample used for the cross sectional properties ($n=88$, according to the Ext Fig 6) or SI Table 2 (several hundred, even though unclear whether the separate male and female listed are included in the $n=770$ listed for the mixed sex sample)?
3. I also recommend to include the modern human comparative sample in the PCA plot of humeral morphometrics (Fig. 2). Fig. 2 is in a somewhat unusual perspective as a partly rotated 3D scatterplot, which makes seeing the effects of the individual principle components difficult. I would recommend to include some regular, 2D scatterplots of PC1 vs. PC2 and PC1 vs. PC3 in the SI. This would also allow to visualize the effects of the individual principal components on the shape (similar to how this is in Fig 3 d), currently this is only done for PC1 in Fig. 2.
4. I was surprised to see that the analyses of in the end very similar data, the outlines of the lower M3 and of the distal humeral cross sections were analyzed using completely different methods, procrustes superimposition and sliding semilandmarks in the case of the humerus, and elliptical Fourier analysis

in the case of the molars. What is the reason for this? I am honestly not sure which of the two is more appropriate (and whether they would give significantly different results), but I am confused why the authors chose these different approaches.

None of these issues subtract from the importance of this paper though, and I don't think they would make a difference with regards to the conclusion that *H. floresiensis* saw a reduction in body size early after its arrival on Flores, and then went through a long, relatively stable period without major changes in body size and only slight changes in dental morphology.

Minor Issues:

Line 430 slice -> slices

Line 555 sizse -> size

Extended data Fig 2.: portions c and d are very pixelated and hard (might be a conversion issue)

Extended data Fig 5.: white line indicating strong axis can not be discerned (might be resolution issue)

Supplementary table 2: Au. prometeus -> prometheus

Reviewer #3:

Remarks to the Author:

This paper reports on new fossils discovered from the site of Mata Menge dated to approximately 700kya. The importance of the paper is threefold: 1) The analyses reinforce the interpretation that, based on morphological characteristics, the Mata Menge remains can be assigned to *H. floresiensis*, 2) that the characteristic small body size is established in this lineage by ca. 700,000 years BP, and 3) the morphological affinities of *H. floresiensis* are most comparable to Javanese *H. erectus* rather than *H. habilis*. The fossils are important, and worthy of the detailed analysis and reporting presented here. I found the analyses of the humerus and new dental remains are thorough and supportive of the interpretations. The paper will make an important contribution to the literature, and I recommend publication with relatively minor revisions that I suggest below.

Page 3 – The new fossils reported are from the upper part of Layer II, the same context as the previously described fossil remains from Mata Menge. While these have been reported before, it would be useful to provide a sentence or two on the dating of the fossils for the current publication, as the age is important to the interpretation of the 'early' reduction in body size. The details of course could be left to other publications, but there needs to be a bit more context on the dating presented here.

Page 4 – line 99 – Developmental age and pathology of the humerus – The use of histomorphology for age determination of the humerus is a good choice. The analyses look convincing in demonstrating that the humerus belongs to a mature adult based on the proportion of secondary osteons, but the images provided in word doc or pdf (Extended data fig 2c and d) are of insufficient resolution for the reader to evaluate this. I assume high quality images will be provided for the publication, but as they appear in the review files the reader has to trust that the authors have identified the osteons and osteon fragments correctly. As I said, I have no reason to question the analysis, but the reader needs to see high resolution images.

Page 4 – line 124 – 'Other characteristics do not contradict the assumption that SOA-MM9 is from a non-pathological adult individual' – Avoid double negatives, change to 'support the interpretation' or similar. Also, the adult status is not an assumption, it is an interpretation, and one that the authors have just quite convincingly demonstrated. Actually, as a further point, this paragraph really attempts to rule out pathological explanations of the small, bodied morphology, so I would consider the age as demonstrated and focus purely on the lack of evidence for pathology in this section. With regards to the section heading (Line 99) the wording "Developmental age and pathology of the humerus (SOA-MM9)" led me to believe that the authors had identified a pathological condition. Since they have not, I would recommend dropping the word 'pathology' or changing it to 'health status'.

Page 5 – Line 141 – How is 19% defined, and why choose 19% over the more standard 30 or 35% locations? I note there is a more detailed description of this on Page 14 lines 425-432, but I think it still requires some clarification. Usually on a complete humerus it is measured from the most distal

point of the trochlea along the shaft, and defined by the total length, which I assume is the case here. At what percentage of length do the authors estimate the distal end of the humerus (as it is preserved) is situated? I recommend some revision to figure 4 below, but it would also be helpful to add some detail to the methods text here. The authors say that it was 'located based on our estimate of its maximum length', but I assume that 19% is also based on the estimated location of the most distal articular point? If so, what is the estimated error range in the location of 19% in mm rather than slices? Conversely, if it's from the preserved distal end, how is the same location estimated for comparative fossil material? Sorry to be pedantic here, but the wording is not absolutely clear. Saying something is located based on an estimated length is one thing, but there is no preserved 'end' of the bone, so there is an additional estimate in the location of that end that also adds error to the positioning section locations and this needs to be absolutely clear in the text.

Page 6 – line 185 – should be estimated rather than calculated

Extended Data Fig. 3 – add 19% level cross-section to figure legend (should be intelligible without reference to text).

Extended Data Fig. 4 – I'm not convinced that listing the slice numbers is useful for the reader. My first question when looking at Fig 4 was to ask where the 19% cross-section was that was used in the previous analysis. This is an unconventional location, so identification of its location would be useful, as would the other section locations mentioned in the text. Can the slice numbers be expressed as distances or percentages of length?

Page 11 Line 325 – should read 'A' or 'The' micro-CT scan'?

Reviewer #4:

Remarks to the Author:

This paper reports new proto-*H. floresiensis* fossils from Mata Menge, Flores. The manuscript is clear, well written and scientifically robust (I am not however a palaeoanthropologist) and the findings are appropriate for the readership of this journal.

The findings are important as they represent the first post-cranial elements and afford new information regarding the likely origins of this hominin and taxonomic relationships. I recommend publication of this paper following attention to the minor comments made below.

Line 54: 'fluvial sandstone' is not correct, change to sandstone of fluvial origin or some such would be more appropriate.

Line 92–97: this is quite a critical observation and I wonder about the implications? Were the bodies interred more rapidly or at least prior to the loss of soft tissues? Were they buried in slackwater sediments that represented significantly lower flow energy? Perhaps if the human fossils are found in the upper part of Layer II this represents a period when flow was decreasing? It feels to me that what this paper is missing is some more detail around the depositional environments and potential taphonomic history of the assemblage. Are there additional data/observations of sediments characteristics in the area where the human fossils were found? If so then some additional contextual information would be appropriate.

Response to referees

Manuscript title: Early evolution of extremely small body size in *Homo floresiensis*

We thank four reviewers for their supportive, encouraging, and helpful comments. Our point-by-point response to each comment is below (the reviewers' comments in blue; our replies in black).

Reviewer #1 (Remarks to the Author):

Kaifu et al. reported three new hominin fossils recovered from SOA-MM and assigned them, along with previously found materials, to *Homo floresiensis*. The more primitive traits and Sangiran affinity directly support the hypothesis that these dwarfed hominins evolved from *H. erectus*, which arrived here in the early Middle Pleistocene, or perhaps even earlier, as hinted by the evidence of artifacts.

Have the authors considered whether the dwarfing process was gradual or a punctuational event, or if it relates to some environmental change?

>> Thank you for this important question. We could narrow down the time range for body size reduction by this study (between ~1.0 and 0.7 Ma) but are not yet ready to answer it. We hope more fossil materials will be discovered from earlier Flores sites to approach this question.

Regarding the cross-sectional geometry, the authors would better provide the original dataset instead of just figures. This will facilitate future comparisons.

>> This is true. New Supplementary Table 4 has been added to report the cross-sectional property data of the SOA-MM9 humerus.

Reviewer #2 (Remarks to the Author):

In this paper, the authors present new finds from Mata Menge, Flores, Indonesia. These add to the previously described material assigned to *Homo floresiensis*, dating to around 700 ka ago, significantly earlier than the eponymous material from Liang Bua (about 60ka). The most important specimen is an adult humerus, which is smaller and more gracile than that of the holotype, LB1. In addition to the humerus, two teeth were also recovered, a maxillary deciduous canine and a lower third molar, both of these are very small. Based on these new discoveries, the authors propose that *floresiensis* was present on Flores since about 700,000 years ago and probably evolved from earlier Asian *Homo erectus*.

Congratulations to the authors on such a well executed and interesting study. These discoveries contribute significantly to our understanding of *Homo floresiensis*, and especially to how and when the remarkably small body size in this species evolved. The authors have provided detailed descriptions of the specimens and of their analyses, and in general, this paper clearly warrants publication due to the importance of the material described within, and the very detailed analyses. There are a few issues though that I think if dealt with would further strengthen the paper:

>>Thank you.

1. I found it rather confusing and hard to understand which comparative sample was used where in the humeral analyses. Even though SI Table 2 lists a very large number of humeri used in the comparative analyses, only a subset of the modern comparative sample was used in the distal humeral GMM analyses. It would be better to be more explicit about this, so the reader doesn't have to hunt for this information in several different places.

>> We added the following texts to resolve this issue.

Line 339 (the Method section): "The modern human samples used for the linear metric analyses are in Supplementary Table 2, while the modern human sample for the geometric morphometric analysis is a mixed-sex sample of adults collected (by J.M. Plavcan) at the Cleveland Museum of Natural History and the Smithsonian National Museum of Natural History (Washington, DC)⁴⁴"

Line 100: "see Supplementary Table 1"

Line 133: "see Supplementary Table 2"

2. Linked to the previous, why was the modern comparative sample for the GMM analyses of the distal humerus so small (n=21, at least according to the Ext. data Fig 7 legend), when compared with the sample used for the cross sectional properties (n=88, according to the Ext Fig 6) or SI Table 2 (several hundred, even though unclear whether the separate male and female listed are included in the n=770 listed for the mixed sex sample)?

>> This is because the original data sets differ between the GMM and other analyses.

3. I also recommend to include the modern human comparative sample in the PCA plot of humeral morphometrics (Fig. 2). Fig. 2 is in a somewhat unusual perspective as a partly rotated 3D scatterplot, which makes seeing the effects of the individual principle components difficult. I would recommend to include some regular, 2D scatterplots of PC1 vs. PC2 and PC1 vs. PC3 in the SI. This would also allow to visualize the effects of the individual principal components on the shape (similar to how this is in Fig 3 d), currently this is only done for

PC1 in Fig. 2.

>> We did not include modern humans from this analysis because our focus was to examine morphological affinities among fossil taxa. For the same reason, we excluded modern humans in our shape analyses (PCA based on the elliptical Fourier analysis) of the molar (Fig. 3). We have presented 2D scatter plots in the new Supplementary Fig. 3.

4. I was surprised to see that the analyses of in the end very similar data, the outlines of the lower M3 and of the distal humeral cross sections were analyzed using completely different methods, Procrustes superimposition and sliding semilandmarks in the case of the humerus, and elliptical Fourier analysis in the case of the molars. What is the reason for this? I am honestly not sure which of the two is more appropriate (and whether they would give significantly different results), but I am confused why the authors chose these different approaches.

>> We employed Elliptical Fourier analysis for the molar due to difficulties in defining its landmarks. However, as this reviewer expects, both analyses should reach similar results as far as each specimen is appropriately oriented.

None of these issues subtract from the importance of this paper though, and I don't think they would make a difference with regards to the conclusion that *H. floresiensis* saw a reduction in body size early after its arrival on Flores, and then went through a long, relatively stable period without major changes in body size and only slight changes in dental morphology.

Minor Issues:

Line 430 slice -> slices >> Corrected

Line 555 size -> size >> Corrected

Extended data Fig 2.: portions c and d are very pixelated and hard (might be a conversion issue) >> The images (now Fig. 3) have been replaced with a better one.

Extended data Fig 5.: white line indicating strong axis can not be discerned (might be resolution issue) >> The image (now Supplementary Fig. 1) has been replaced with a better one.

Supplementary table 2: *Au. prometeus* -> *prometheus* >> Corrected

Reviewer #3 (Remarks to the Author):

This paper reports on new fossils discovered from the site of Mata Menge dated to approximately 700kya. The importance of the paper is threefold: 1) The analyses reinforce

the interpretation that, based on morphological characteristics, the Mata Menge remains can be assigned to *H. floresiensis*, 2) that the characteristic small body size is established in this lineage by ca. 700,000 years BP, and 3) the morphological affinities of *H. floresiensis* are most comparable to Javanese *H. erectus* rather than *H. habilis*. The fossils are important, and worthy of the detailed analysis and reporting presented here. I found the analyses of the humerus and new dental remains are thorough and supportive of the interpretations. The paper will make an important contribution to the literature, and I recommend publication with relatively minor revisions that I suggest below.

Page 3 – The new fossils reported are from the upper part of Layer II, the same context as the previously described fossil remains from Mata Menge. While these have been reported before, it would be useful to provide a sentence or two on the dating of the fossils for the current publication, as the age is important to the interpretation of the ‘early’ reduction in body size. The details of course could be left to other publications, but there needs to be a bit more context on the dating presented here.

>>We included this information in the revised “Context and geological age” section of the Results.

Page 4 – line 99 – Developmental age and pathology of the humerus – The use of histomorphology for age determination of the humerus is a good choice. The analyses look convincing in demonstrating that the humerus belongs to a mature adult based on the proportion of secondary osteons, but the images provided in word doc or pdf (Extended data fig 2c and d) are of insufficient resolution for the reader to evaluate this. I assume high quality images will be provided for the publication, but as they appear in the review files the reader has to trust that the authors have identified the osteons and osteon fragments correctly. As I said, I have no reason to question the analysis, but the reader needs to see high resolution images.

>>We apologize for the inconvenience. We submitted high-resolutions images.

Page 4 – line 124 – ‘Other characteristics do not contradict the assumption that SOA-MM9 is from a non-pathological adult individual’ – Avoid double negatives, change to ‘support the interpretation’ or similar. Also, the adult status is not an assumption, it is an interpretation, and one that the authors have just quite convincingly demonstrated. Actually, as a further point, this paragraph really attempts to rule out pathological explanations of the small, bodied morphology, so I would consider the age as demonstrated and focus purely on the lack of evidence for pathology in this section. With regards to the section heading (Line 99) the

wording “Developmental age and pathology of the humerus (SOA-MM9)” led me to believe that the authors had identified a pathological condition. Since they have not, I would recommend dropping the word ‘pathology’ or changing it to ‘health status’.

>> We appreciate these helpful comments. Because we have not (cannot) examined the health status, we simplified the subheading as “Developmental age of the humerus (SOA-MM9)”.

Page 5 – Line 141 – How is 19% defined, and why choose 19% over the more standard 30 or 35% locations? I note there is a more detailed description of this on Page 14 lines 425-432, but I think it still requires some clarification. Usually on a complete humerus it is measured from the most distal point of the trochlea along the shaft, and defined by the total length, which I assume is the case here. At what percentage of length do the authors estimate the distal end of the humerus (as it is preserved) is situated? I recommend some revision to figure 4 below, but it would also be helpful to add some detail to the methods text here. The authors say that it was ‘located based on our estimate of its maximum length’, but I assume that 19% is also based on the estimated location of the most distal articular point? If so, what is the estimated error range in the location of 19% in mm rather than slices? Conversely, if it’s from the preserved distal end, how is the same location estimated for comparative fossil material? Sorry to be pedantic here, but the wording is not absolutely clear. Saying something is located based on an estimated length is one thing, but there is no preserved ‘end’ of the bone, so there is an additional estimate in the location of that end that also adds error to the positioning section locations and this needs to be absolutely clear in the text.

>>> Previous studies of fossil hominin humeri have demonstrated taxonomic utility of the cross-sectional shape of the distal diaphysis sampled at ~19% of total (biomechanical) humerus length from the distal end^{29,44,46,54-56}. The ~19% level of SOA-MM9 was located based on our estimate of its maximum length (211–220 mm), which was converted to the biomechanical humeral length using the ratio between the two (the former is 1.08% longer on average in our mixed-sex, prehistoric modern human (Jomon) sample: N = 88). The 19% level thus located is within the CT slice nos. 573–639. Therefore, we chose three slices, nos. 573, 607 (best estimate), and 639 for the present analysis. Two-dimensional coordinates were collected from all three sections of SOA-M9 following the procedure described by Lague⁴⁴ (i.e., two Type 2 landmarks on the medial and lateral extremes of the specimen and 58 sliding semilandmarks on the anterior and posterior surfaces). Raw landmark configurations were superimposed into the same shape space using orthogonal least-squares generalized Procrustes (GPA) superimposition⁵⁷. GPA was performed using tpsRelw software⁵⁸ and semilandmarks were allowed to slide along the diaphyseal outline using the

criterion of minimized bending energy⁵⁹. Subsequent to GPA, morphometric relationships were assessed with the use of Procrustes distances (Dp) as a measure of shape dissimilarity⁶⁰ and principal component analysis (PCA) as a means of visual summary (via ordination) of shape variation among the individual specimens.

Page 6 – line 185 – should be estimated rather than calculated

>> Corrected

Extended Data Fig. 3 – add 19% level cross-section to figure legend (should be intelligible without reference to text).

>> Corrected

Extended Data Fig. 4 – I'm not convinced that listing the slice numbers is useful for the reader. My first question when looking at Fig 4 was to ask where the 19% cross-section was that was used in the previous analysis. This is an unconventional location, so identification of its location would be useful, as would the other section locations mentioned in the text. Can the slice numbers be expressed as distances or percentages of length?

>> We added the level of our estimated 19% level in Fig. 5 (former Extended Data Fig. 4). These slice numbers are identification number for each CT slice. We do not replace them with distances or percentages of length because the latter are estimates.

Page 11 Line 325 – should read 'A' or 'The' micro-CT scan'?

>> Corrected

Reviewer #4 (Remarks to the Author):

This paper reports new proto-H. floresiensis fossils from Mata Menge, Flores. The manuscript is clear, well written and scientifically robust (I am not however a palaeoanthropologist) and the findings are appropriate for the readership of this journal.

The findings are important as they represent the first post-cranial elements and afford new information regarding the likely origins of this hominin and taxonomic relationships. I recommend publication of this paper following attention to the minor comments made below.

Line 54: 'fluvial sandstone' is not correct, change to sandstone of fluvial origin or some such would be more appropriate.

>>Corrected

Line 92–97: this is quite a critical observation and I wonder about the implications? Were the bodies interred more rapidly or at least prior to the loss of soft tissues? Were they buried in slackwater sediments that represented significantly lower flow energy? Perhaps if the human fossils are found in the upper part of Layer II this represents a period when flow was decreasing? It feels to me that what this paper is missing is some more detail around the depositional environments and potential taphonomic history of the assemblage. Are there additional data/observations of sediments characteristics in the area where the human fossils were found? If so then some additional contextual information would be appropriate.

>>To address what we can infer about the taphonomy of these fossils, we included brief sentences in the main text (at the end of the “Context and geological age”) and added the new Supplementary Note 1 in Supplementary Information.

Reviewers' Comments:

Reviewer #2:

Remarks to the Author:

The authors took all of my suggestions into account, and answered my questions in great detail. This is a great paper, and I am happy to see it published.

Bence Viola

Reviewer #3:

Remarks to the Author:

I have reviewed the revised manuscript and the revisions made in response to my previous recommendations. The authors have fully addressed the suggestions that I made previously, and the revised manuscript is polished and ready for publication.

Reviewer #4:

Remarks to the Author:

My feedback was limited to a few minor comments, all of which have been addressed satisfactorily, many thanks and congratulations